REGISTERED REPORT

# Registered report: Tumour micro-environment elicits innate resistance to RAF inhibitors through HGF secretion

**David Blum[1], Samuel LaBarge[2], Reproducibility Project: Cancer Biology\*†**

[1]Bioexpression and Fermentation Facility, University of Georgia, Athens, United States; [2]City of Hope, Duarte, United States

**Abstract** The Reproducibility Project: Cancer Biology seeks to address growing concerns about reproducibility in scientific research by conducting replications of 50 papers in the field of cancer biology published between 2010 and 2012. This Registered Report describes the proposed replication plan of key experiments from 'Tumour micro-environment elicits innate resistance to RAF inhibitors through HGF secretion' by Straussman and colleagues, published in Nature in 2012 (*Straussman et al., 2012*). The key experiments being replicated in this study are from Figure 2A, C, and D (and Supplemental Figure 11) and Figure 4C (and Supplemental Figure 19) (*Straussman et al., 2012*). Figure 2 demonstrates resistance to drug sensitivity conferred by co-culture with some stromal cell lines and identifies the secreted factor responsible as HGF. In Figure 4, Straussman and colleagues show that blocking the HGF receptor MET abrogates HGF's rescue of drug sensitivity. The Reproducibility Project: Cancer Biology is a collaboration between the Center for Open Science and Science Exchange, and the results of the replications will be published by *eLife*.

**\*For correspondence:** fraser@scienceexchange.com

**Group author details**
†Reproducibility Project: Cancer Biology
See page 27

**Reviewing editor**: Charles L Sawyers, Memorial Sloan-Kettering Cancer Center, United States

## Introduction

Resistance to oncoprotein-targeted chemotherapy is a common occurrence during cancer treatment and identifying the mechanisms of resistance is important in improving treatment options. Specifically, BRAF-mutant melanomas, which show an initial response to RAF inhibitors, usually become resistant to the therapy (*Nickoloff and Vande Woude, 2012*). The identification of stroma-mediated resistance in BRAF-mutant melanomas, through the secretion of hepatocyte growth factor (HGF), therefore indicates a potential therapeutic strategy through combination treatment of RAF inhibitors and inhibition of the HGF activated pathway (*Straussman et al., 2012*). This report is the first to identify paracrine HGF as a potential mechanism for the development of drug resistance (*Ghiso and Giordano, 2013*; *Glaire et al., 2012*).

In Figure 2A of their paper, Straussman and colleagues tested the effect of fibroblast-conditioned medium on the proliferation of BRAF-mutant melanoma cells grown in the presence of the BRAF inhibitor PLX4720. Using a cell proliferation assay, they reported that fibroblast-conditioned medium rescued BRAF-mutant melanoma cells from PLX4720 sensitivity, which indicated that a secreted factor was involved. This was a key finding demonstrating that the stromal environment of the tumor cells could mediate their response to drug treatment. This experiment will be replicated in Protocol 3.

Straussman and colleagues went on to identify the secreted factor responsible for acquired drug resistance as HGF. In Figure 2C, they demonstrated that treating melanoma cell lines with PLX4720 in combination with increasing concentrations of exogenous HGF increased proliferation as compared to treatment with drug alone. This finding showed a similar effect to treatment with conditioned media from stromal cells that secrete HGF (see Figure 2A) and supported the hypothesis that HGF is the growth factor responsible for rescuing melanoma cells from drug sensitivity. This experiment will be replicated in Protocol 4.

Straussman and colleagues demonstrated that the HGF-mediated rescue of melanoma cells from drug sensitivity was mediated through HGF's cognate receptor tyrosine kinase MET by treating melanoma cell lines co-cultured with stromal cell lines in the presence of PLX4720 with the MET inhibitor crizotinib, as shown in Figure 2D and Supplemental Figure 11. Treatment with crizotinib reduced the increase in proliferation due to co-culture with an HGF-secreting stromal cell line. This experiment provided further support for the hypothesis that HGF was responsible for rescue from drug sensitivity and also provided evidence that that rescue was MET dependent. This experiment is replicated in Protocol 5.

Lastly, Straussman and colleagues reported sustained activation of both ERK and AKT in HGF-treated melanoma cells during BRAF inhibition and to a lesser extent MEK inhibition, as shown in Figure 4C and Supplemental Figure 19 by Western blot. This confirmed activation of pro-survival pathways in response to HGF treatment even in the presence of PLX4720. These experiments are replicated in Protocol 6.

To date, a direct replication has been reported; Lezcano and colleagues (*Lezcano et al., 2014*) published a replication of Figure 3 of Straussman et al. Nature 2013, wherein Straussman and colleagues evaluated HGF expression in patient-derived primary melanoma samples and observed a negative correlation between expression of HGF and response to therapy (*Straussman et al., 2012*). While Lezcano and colleagues' replication also detected the presence of HGF in human melanoma tumor cells and stromal cells with increased expression at disease progression, they did not identify a statistically significant correlation between HGF expression and clinical outcome (*Lezcano et al., 2014*). While both of the studies come to different conclusions about the association of stromal HGF and clinical outcome, the 95% confidence intervals of the standardized measure of the effect (Cohen's *d*) for each study substantially overlap. A study published around the same time as the work of Straussman and colleagues supports the negative association between HGF and clinical response to RAF inhibitor treatments through an analysis of HGF levels in patient plasma samples (*Wilson et al., 2012*).

In other systems, additional labs have observed a similar role for HGF in acquired drug resistance. Caenepeel and colleagues reported that HGF rescued melanoma cell lines, notably SK-MEL-5, from BRAF or MEK inhibition using vemurafenib (an analogue of PLX4720) or PD0325901, respectively, and the rescue was attenuated by MET inhibition (*Caenepeel et al., 2013*). Nakagawa and colleagues observed that tumor-secreted (not stromal secreted) HGF could induce resistance to the VEGFR inhibitor lenvatinib, and that this resistance could be overcome by co-treatment with golvatinib, a MET inhibitor (*Nakagawa et al., 2014*). Etnyre and colleagues reported that c-MET and BRAF inhibitors had synergistic inhibitory effects when exposed in combination to melanoma cell lines (*Etnyre et al., 2013*). Casbas-Hernandez and colleagues co-cultured MCF10 cells with immortalized mammoplasty derived fibroblasts and observed a correlation between the levels of fibroblast-secreted HGF and the differentiation of the MCF10 cells towards a ductal carcinoma phenotype. They also observed a correlation between HGF expression and the more invasive basal-like tumors as opposed to the less invasive luminal tumors (*Casbas-Hernandez et al., 2013*). HGF is also being evaluated as a potential biomarker to indicate potential treatment choices (*Penuel et al., 2013*; *Xie et al., 2013*).

## Materials and methods

Unless otherwise noted, all protocol information was derived from the original paper, references from the original paper, or information obtained directly from the authors.

### Protocol 1: determining the range of detection of the replicating lab's plate reader

This is a general protocol that determines the range of detection of the plate reader. Because the plate reader in use by the replicating lab is different than the plate reader used in the original study, we are determining what the range of detection is for the replicating lab's plate reader.

### Sampling

- SK-MEL-5

    1. 8000 cells/well x 4 replicates
    2. 4000 cells/well x 4 replicates
    3. 2000 cells/well x 4 replicates
    4. 1000 cells/well x 4 replicates
    5. 500 cells/well x 4 replicates

6. 250 cells/well x 4 replicates
7. 125 cells/well x 4 replicates
8. 62.5 cells/well x 4 replicates
9. 31.25 cells/well x 4 replicates

- The experiment is done a total of once.

## Materials and reagents
- Reagents that are different from ones originally used are noted with an asterisk (*).

| Reagent | Type | Manufacturer | Catalog # | Comments |
|---|---|---|---|---|
| pLEX-TRC206 SK-MEL-5 | Cells | Original authors | n/a | Engineered to express GFP |
| Synergy HT Microplate Reader* | Equipment | Bio-Tek | | Original equipment used: Molecular Devices SpectraMax M5e Microplate Reader |
| 384-well clear-bottomed plates | Material | Corning | 3712 | |
| Phenol red free DMEM* | Medium | Sigma-Aldrich | D1145 | Original unspecified. |
| Sodium pyruvate solution* | Reagent | Sigma-Aldrich | S8636 | This formulation of DMEM does not contain L-glutamine or sodium pyruvate, so these will be supplemented to the medium. |
| FBS* | Reagent | Sigma-Aldrich | F4135 | Original unspecified |
| 100X Pen–Strep–Glut* | Reagent | Sigma-Aldrich | G1146 | Original from Invitrogen (15,140-122) |
| Puromycin dihydrochloride | Reagent | Sigma-Aldrich | P9620 | Original unspecified |
| Biomek FX* | Equipment | Beckman Coulter | | Communicated by authors. Original from Thermo Scientific (Combi reagent dispenser) |

## Procedure

1. Seed 4 wells of a 384-well clear-bottom plate with 8000 cells/well all the way to 31.25 cells/well (serial 1:2 dilutions) with pLex-TRC206 SK-MEL-5 cells in 60 µl per well using phenol red free medium using an automated workstation.

Note: all cells will be sent for mycoplasma testing and STR profiling.
Note: ensure at least 85% of SK-MEL-5 cells are GFP-positive before start of the experiment. Cells can be enriched using FACS or puromycin (0.5–2 µg/ml), however do not grow cells under antibiotic selection on a regular basis.

   A. Total wells seeded = 36
   B. Medium for assay: phenol red free DMEM supplemented with 1 mM sodium pyruvate, 10% FBS, and 1X Pen–Strep–Glut.
   C. Fill wells with 60 µl/well of clear media in at least 2 rows and 2 columns around wells that are being included in the experiment.

2. The next day after seeding, read GFP fluorescence (Synergy HT Microplate Reader).

   A. Subtract the average reading from media-only wells from the wells with cells.

## Deliverables

- Data to be collected:

   1. Raw GFP fluorescence readings.
   2. Graph of GFP fluorescence readings vs cell number.

## Confirmatory analysis plan

- Statistical Analysis:

  1. Coefficient of determination of data values.

## Known differences from the original study

- Synergy HT Microplate Reader used instead of Molecular Devices SpectraMax M5e Microplate Reader—both can detect GFP fluorescence and the Synergy HT Microplate Reader will be evaluated for sensitivity of detection (Protocol 1) and to determine if the gradient is similar to the original study ($\leq$5%) (Protocol 2).

## Provisions for quality control

This protocol will ensure that the replicating lab's plate reader is comparable to the original lab's plate reader.

- A lab from the Science Exchange network with extensive experience in conducting cell viability assays will perform these experiments.
- All cells will be sent for STR profiling to confirm identity and mycoplasma testing to confirm the lack of mycoplasma contamination.
- SK-MEL-5 cells will be confirmed to have at least 85% of the cells GFP-positive before the start of the experiment.

## Protocol 2: determining the detection variability of the replicating lab's plate reader

This is a general protocol that determines the variability in detection of the plate reader. Because the plate reader in use by the replicating lab is different than the plate reader used in the original study, we are determining what the variability of detection is for the replicating lab's plate reader.

## Sampling

- SK-MEL-5:

  1. 2000 cells/well x 384 replicates

- Experiment will be done a total of once.

## Materials and reagents

• Reagents that are different from ones originally used are noted with an asterisk (*).

| Reagent | Type | Manufacturer | Catalog # | Comments |
|---|---|---|---|---|
| pLEX-TRC206 SK-MEL-5 | Cells | Original authors | n/a | Engineered to express GFP |
| Synergy HT Microplate Reader* | Equipment | Bio-Tek | | Original equipment used: Molecular Devices SpectraMax M5e Microplate Reader |
| 384-well clear-bottomed plates | Material | Corning | 3712 | |
| Phenol red free DMEM* | Medium | Sigma-Aldrich | D1145 | Original unspecified. This formulation of DMEM does not contain L-glutamine or sodium pyruvate, so these will be supplemented to the medium. |
| Sodium pyruvate solution* | Reagent | Sigma-Aldrich | S8636 | |
| FBS* | Reagent | Sigma-Aldrich | F4135 | Original unspecified |
| 100X Pen–Strep–Glut* | Reagent | Sigma-Aldrich | G1146 | Original from Invitrogen (15,140-122) |
| Puromycin dihydrochloride | Reagent | Sigma-Aldrich | P9620 | Original unspecified |
| Biomek FX | Equipment | Beckman Coulter | | Communicated by authors. Original from Thermo Scientific (Combi reagent dispenser) |

## Procedure

1. Seed all wells of a 384 -well clear-bottom plate with 2000 pLex-TRC206 SK-MEL-5 cells (provided by authors) in 60 μl per well using phenol red free medium using an automated workstation.

Note: all cells will be sent for mycoplasma testing and STR profiling.
Note: ensure at least 85% of SK-MEL-5 cells are GFP-positive before start of the experiment. Cells can be enriched using FACS or antibiotics, however do not grow cells under antibiotic selection on a regular basis.

 A. Medium for assay: phenol red free DMEM supplemented with 1 mM sodium pyruvate, 10% FBS, and 1× Pen–Strep–Glut.
 B. Fill wells with 60 μl/well of clear media in at least 2 rows and 2 columns around wells that are being included in the experiment.

2. The next day after seeding, read GFP fluorescence (Synergy HT Microplate Reader).

 A. Subtract the average reading from media only wells from the wells with cells.

## Deliverables

- Data to be collected:

  1. Raw GFP fluorescence readings.
  2. Difference of each individual well and the average reading across the plate.

## Confirmatory analysis plan

- Statistical Analysis:

  1. Standard deviation of data values.

## Known differences from the original study

- Synergy HT Microplate Reader used instead of Molecular Devices SpectraMax M5e Microplate Reader—both can detect GFP fluorescence and the Synergy HT Microplate Reader will be evaluated for sensitivity of detection (Protocol 1) and to determine if the gradient is similar to the original study (≤5%) (Protocol 2).

## Provisions for quality control
This protocol will ensure that the replicating lab's plate reader is comparable to the original lab's plate reader.

- A lab from the Science Exchange network with extensive experience in conducting cell viability assays will perform these experiments.
- All cells will be sent for STR profiling to confirm identity and mycoplasma testing to confirm the lack of mycoplasma contamination.
- SK-MEL-5 cells will be confirmed to have at least 85% of the cells GFP-positive before the start of the experiment.

## Protocol 3: co-culture proliferation assay
This protocol outlines how to culture melanoma cell lines with conditioned medium from three stromal cell lines with or without the RAF inhibitor PLX4720 to analyze cell proliferation rates, as is described in Figure 2A.

## Sampling

- Experiment to be repeated a total of 4 times for a minimum power of 81%.

  1. See Power calculations section for details

- Each experiment has six conditions to be run in quadruplicate per experiment:

  1. SK-MEL-5 untreated control [additional control]
  2. SK-MEL-5 vehicle (DMSO) control
  3. SK-MEL-5 treated with 2 µM PLX4720 and with unconditioned medium
  4. SK-MEL-5 treated with 2 µM PLX4720 and with conditioned medium from CCD-1090Sk cells that do not secrete HGF
  5. SK-MEL-5 treated with 2 µM PLX4720 and with conditioned medium from PC60163A1 cells that do secrete HGF
  6. SK-MEL-5 treated with 2 µM PLX4720 and with conditioned medium from LL 86 cells that do secrete HGF

## Materials and reagents

- Reagents that are different from ones originally used are noted with an asterisk (*).

| Reagent | Type | Manufacturer | Catalog # | Comments |
|---|---|---|---|---|
| LL 86 cells | Cells | Original authors | n/a | Stromal cell line that secretes HGF |
| PC60163A1 | Cells | Original authors | n/a | Stromal cell line that secretes HGF |
| CCD-1090Sk cells | Cells | Original authors | n/a | Stromal cell line that does not secrete HGF |
| pLEX-TRC206 SK-MEL-5 | Cells | Original authors | n/a | Engineered to express GFP |
| Synergy HT Microplate Reader* | Equipment | Bio-Tek | | Original equipment used: Molecular Devices SpectraMax M5e Microplate Reader |
| Pathway 435 Bioimager | Equipment | BD Biosciences | | Original equipment used: Zeiss Axio Observer.Z1 |
| 384-well clear-bottomed plates | Material | Corning | 3712 | |
| 10 cm tissue culture plates* | Materials | Corning | 430167 | Original unspecified |
| 0.45 µm syringe filter | Materials | Sigma-Aldrich | Z355518 | Original unspecified |
| 10 ml syringe | Materials | Sigma-Aldrich | Z116874 | Original unspecified |
| Phenol red free DMEM* | Medium | Sigma-Aldrich | D1145 | Original unspecified. This formulation of DMEM does not contain L-glutamine or sodium pyruvate, so these will be supplemented to the medium. |
| Sodium pyruvate solution* | Reagent | Sigma-Aldrich | S8636 | |
| FBS* | Reagent | Sigma-Aldrich | F4135 | Original unspecified |
| 100X Pen–Strep–Glut* | Reagent | Sigma-Aldrich | G1146 | Original from Invitrogen (15,140-122) |
| PLX4720 | Drug | Chemietek | CT-P4720 | |
| DMSO* | Reagent | Sigma-Aldrich | D8418 | Original unspecified |
| Biomek FX | Equipment | Beckman Coulter | | Communicated by authors. Original from Thermo Scientific (Combi reagent dispenser) and CyBio robotic liquid handler. |

## Procedure

1. Prepare Pre-Conditioned Medium (PCM); fresh PCM must be prepared the same day it is used in the treatment of SK-MEL-5 cells; this step is repeated three times to ensure fresh PCM is available on the needed day:

   A. Three days before the PCM is needed, seed 3 × 10 cm tissue culture plates with $0.5 \times 10^6$ LL 86 cells each, 3 × 10 cm tissue culture plates with $1 \times 10^6$ PC60163A1 cells each, and 3 × 10 cm tissue culture plates with $2 \times 10^6$ CCD-1090Sk cells each (9 plates total) in 10 ml of phenol red free medium each and grow for 3 days.

   B. 3 days after seeding, collect the medium from each cell line using the plate closest to 80–90% confluent.

      i. 75–95% confluency can be used.

 C. Filter through 0.45 μm syringe filter with a 10 ml syringe and dilute filtered PCM 1:1 in fresh phenol red free medium. Total volume = 20 ml.

 i. Use the same day.
 ii. Do not dilute for day 0 of treatment (these wells will already have 20 μl of media in them).

2. On day 0, seed 120 wells of a 384-well clear-bottom plate with 1900 pLex-TRC206 SK-MEL-5 cells in 20 μl per well using phenol red free medium using an automated workstation.

 Note:

 1. All cells will be sent for mycoplasma testing and STR profiling.
 2. Ensure at least 85% of SK-MEL-5 cells are GFP-positive before start of the experiment. Cells can be enriched using FACS or antibiotics, however do not grow cells under antibiotic selection on a regular basis.
 3. Do not exceed a rate of 5–10 μl/s and do not let the tip end closer than 1 mm to the well bottom.

 A. Fill wells with 50 μl/well of media in at least 2 rows and 2 columns around wells that are being included in the experiment.

 i. Medium for assay: phenol red free DMEM supplemented with 1 mM sodium pyruvate, 10% FBS, and 1X Pen–Strep–Glut.

 B. To wells in step A, add 20 μl of fresh undiluted PCM from appropriate stromal cells generated as described in step 1 (see Sampling section for Cohorts) or phenol red free medium alone (Cohort 1).

3. On day 1 after seeding, read GFP fluorescence (Synergy HT Microplate Reader).

 A. Subtract the average reading from media-only wells from the wells with cells.

4. After reading GFP fluorescence, refresh media and add drug using an automated workstation.

 A. Change the medium for each cohort to 40 μl fresh diluted PCM from appropriate stromal cell lines generated as described in step 1 or phenol red free medium alone.
 B. Within each cohort, add 10 μl of 5X PLX4720, DMSO dilution, or 10 μl phenol red free medium to each appropriate well to bring the final volume per well up to 50 μl.

 i. 5X PLX4720: make up stocks of 10 mM PLX4720 in DMSO, then dilute 1:1000 in media to make up 10 μM PLX4720. This is a 5× stock.
 ii. DMSO dilution: dilute 1 μl DMSO with 999 μl media. Add 10 μl of this mix to DMSO wells.

 1. These dilutions in media prevent toxicity from excess DMSO.

5. On day 4 after seeding, read GFP fluorescence.

 A. Subtract the average reading from media-only wells from the wells with cells.

6. After reading GFP fluorescence, change the medium in appropriate wells to 40 μl fresh diluted PCM from appropriate stromal cell lines generated as described in step 1 or phenol red free medium alone using an automated workstation.

 A. Add 10 μl of 5X PLX4720, DMSO dilution, or 10 μl phenol red free medium to each appropriate well to bring the final volume per well up to 50 μl.

 i. 5X PLX4720: make up stocks of 10 mM PLX4720 in DMSO, then dilute 1:1000 in media to make up 10 μM PLX4720. This is a 5× stock.
 ii. DMSO dilution: dilute 1 μl DMSO with 999 μl media. Add 10 μl of this mix to DMSO wells.

7. On day 7 after seeding, read GFP fluorescence and document bright-field and GFP images (BD, Pathway 435 Bioimager).

 A. Subtract the average reading from media-only wells from the wells with cells.

8. Data analysis:

A. Remove background fluorescence by subtracting the average reading from media-only wells from the wells with cells for each plate reading.
B. Subtract the readings of day 1 from the other plates (day 4 and day 7) for the same wells.
C. Average the quadruplicates.
D. Calculate the effect of PLX4720 in the presence or absence of conditioned media by normalizing the number of cells after 7 days of treatment (as measured by GFP fluorescence) to the number of cells present in the SK-MEL-5 vehicle control condition.

9. Repeat experiment independently three additional times.

## Deliverables

- Data to be collected:

  1. Raw GFP fluorescence readings from days 1, 4, and 7.
  2. Normalized fluorescence proliferation data.
  3. Fluorescent and bright-field micrographs of cells from day 7.
  4. Bar chart of relative proliferation as a % of untreated control for all conditions. (Use data from Day 7–Day 1 background) (Compare to Figure 2A)
  5. A semi-logarithmic graph of proliferation (log) vs time (linear) over three time points after seeding.

## Confirmatory analysis Plan

- Statistical analysis of the replication data:

  A. One-way ANOVA comparing the proliferation of PLX4720-treated cells cultured with unconditioned medium, CCD-1090Sk conditioned medium, LL 86 conditioned medium, or PC60163A1 conditioned medium.

    1. Planned comparisons with the Bonferroni correction:

       - Unconditioned medium to PC60163A1 conditioned medium
       - Unconditioned medium to LL 86 conditioned medium
       - CCD-1090Sk to PC60163A1 conditioned medium
       - CCD-1090Sk to LL 86 conditioned medium

 Meta-analysis of original and replication attempt effect sizes:

  A. Compare the effect sizes of the original data to the replication data and use a meta-analytic approach to combine the original and replication effects, which will be presented as a forest plot.

## Known differences from the original study

- The replication will only use one of the three melanoma cell lines used by the original authors, the SK-MEL-5 cell line. The replication will exclude SK-MEL-28 and G-361 cells.
- The replication will include an additional control, untreated SK-MEL-5 cells in addition to the vehicle (DMSO) treated SK-MEL-5 cells used in the original study.
- A Synergy HT Microplate Reader will be used instead of a Molecular Devices SpectraMax M5e Microplate Reader—both can detect GFP fluorescence and the Synergy HT Microplate Reader will be evaluated for range of detection (Protocol 1) and detection variability (Protocol 2)
- A BD Pathway 435 Bioimager used instead of a Zeiss Axio Observer.Z1—both are fluorescence microscopes with high-throughput screening capabilities.
- The replicating lab does not have a ViCell XR cell viability counter, and thus will seed a larger number of cells per well (1900 instead of 1700 cells/well).

## Provisions for quality control
All data obtained from the experiment—raw data, data analysis, control data, and quality control data - will be made publicly available, either in the published manuscript or as an open access dataset available on the Open Science Framework (https://osf.io/p4lzc/).

- A lab from the Science Exchange network with extensive experience in conducting cell viability assays will perform these experiments.
- All cells will be sent for STR profiling to confirm identity and mycoplasma testing to confirm the lack of mycoplasma contamination.
- SK-MEL-5 cells will be confirmed to have at least 85% of the cells GFP-positive before the start of the experiment.

## Protocol 4: recombinant HGF proliferation assay

This protocol assesses changes in proliferation when melanoma cells are treated with the RAF inhibitor PLK4720 with or without HGF, as is described in Figure 2C. The cells are also treated with a MEK inhibitor, PD184352.

### Sampling

- Experiment to be repeated a total of three times for a final power of 99%.

    1. See Power calculations section for details

- Each experiment has 12 conditions to be done in quadruplicate per experiment:

    1. SK-MEL-5 untreated control [additional control]
    2. SK-MEL-5 vehicle (DMSO) control
    3. SK-MEL-5 2 µM PLX4720 + 0 ng/ml HGF
    4. SK-MEL-5 2 µM PLX4720 + 6.25 ng/ml HGF
    5. SK-MEL-5 2 µM PLX4720 + 12.5 ng/ml HGF
    6. SK-MEL-5 2 µM PLX4720 + 25 ng/ml HGF
    7. SK-MEL-5 2 µM PLX4720 + 50 ng/ml HGF
    8. SK-MEL-5 1 µM PD184352 + 0 ng/ml HGF
    9. SK-MEL-5 1 µM PD184352 + 6.25 ng/ml HGF
    10. SK-MEL-5 1 µM PD184352 + 12.5 ng/ml HGF
    11. SK-MEL-5 1 µM PD184352 + 25 ng/ml HGF
    12. SK-MEL-5 1 µM PD184352 + 50 ng/ml HGF

### Materials and reagents

• Reagents that are different from the ones originally used are noted with an asterisk (*).

| Reagent | Type | Manufacturer | Catalog # | Comments |
|---|---|---|---|---|
| pLEX-TRC206 SK-MEL-5 | Cells | Original authors | n/a | Engineered to express GFP |
| PLX4720 | Drug | Chemietek | CT-P4720 | |
| PD184352 | Drug | Santa Cruz | sc-202759A | MEK inhibitor |
| 384-well clear-bottomed plates | Material | Corning | 3712 | |
| 0.45 µm syringe filter | Materials | Sigma-Aldrich | Z355518 | Original unspecified |
| 10 ml syringe | Materials | Sigma-Aldrich | Z116874 | Original unspecified |
| Phenol red free DMEM* | Medium | Sigma-Aldrich | D1145 | Original unspecified. This formulation of DMEM does not contain L-glutamine or sodium pyruvate, so these will be supplemented to the medium. |
| Sodium pyruvate solution* | Reagent | Sigma-Aldrich | S8636 | |
| FBS* | Reagent | Sigma-Aldrich | F4135 | Original unspecified |
| 100X Pen–Strep–Glut* | Reagent | Sigma-Aldrich | G1146 | Original from Invitrogen (15,140-122) |
| HGF | Reagent | Sigma-Aldrich | H5791 | Original from RayBiotech (228-10,702-2) |
| DMSO* | Reagent | Sigma-Aldrich | D8418 | Original unspecified |
| Synergy HT Microplate Reader* | Equipment | Bio-Tek | | Original equipment used: Molecular Devices SpectraMaxM5e Microplate Reader |
| Biomek FX | Equipment | Beckman Coulter | | Communicated by authors. Original from Thermo Scientific (Combi reagent dispenser) and CyBio robotic liquid handler. |

## Procedure

1. On day 0, seed 48 wells of a 384-well clear-bottom plate with 2800 pLex-TRC206 SK-MEL-5 cells in 40 µl of phenol red free medium each using an automated workstation.
Note:

   1. All cells will be sent for mycoplasma testing and STR profiling.
   2. Ensure at least 85% of SK-MEL-5 cells are GFP-positive before start of the experiment. Cells can be enriched using FACS or antibiotics, however do not grow cells under antibiotic selection on a regular basis.
   3. Do not exceed a rate of 5–10 µl/s and do not let the tip end closer than 1 mm to the well bottom.
   A. Fill wells with 60 µl/well of clear media in at least 2 rows and 2 columns around wells that are being included in the experiment.
   B. Medium of all cell lines for assay: phenol red free DMEM supplemented with 1 mM sodium pyruvate, 10% FBS, and 1× Pen–Strep–Glut.

2. On day 1 after seeding, read GFP fluorescence (Synergy HT Microplate Reader).

   A. Subtract the average reading from media-only wells from the wells with cells.

3. After reading GFP fluorescence, add to the appropriate wells 10 µl 6X HGF or phenol red free medium alone. Then add to the appropriate wells the following: 10 µl 6X PLX4720, 10 µl 6X PD184352, 10 µl DMSO dilution, or 10 µl phenol red free medium alone.

   A. 6X HGF: make up stocks of 100 µg/ml HGF, then dilute accordingly to make 6X working concentrations of each required HGF dilution.
   B. 6X PLX4720: make up stocks of 12 mM PLX4720 in DMSO, then dilute 1:1000 in media to make up 12 µM PLX4720 for use at 6X for the assay to avoid excessive DMSO toxicity.
   C. 6X PD184352: make up stocks of 6 mM PD184352 in DMSO, then dilute 1:1000 in media to make up 6 µM PD184352 for use at 6X for the assay to avoid excessive DMSO toxicity.
   D. DMSO dilution: dilute 1 µl DMSO with 999 µl media. Add 10 µl of this mix to DMSO dilution wells.

      A. These media dilutions are to avoid toxicity from excessive DMSO.

4. On day 4 after seeding, read GFP fluorescence.

   A. Subtract the average reading from media-only wells from the wells with cells.

5. After reading GFP fluorescence, change the medium in all wells to 40 µl fresh phenol red free medium using an automated workstation. Then add to the appropriate wells 10 µl 6X HGF or phenol red free medium alone. Then add to the appropriate wells the following: 10 µl 6X PLX4720, 10 µl 6X PD184352, 10 µl DMSO dilution, or 10 µl phenol red free medium alone.

   A. 6X HGF: make up stocks of 100 µg/ml HGF, then dilute accordingly to make 6X working concentrations of each required HGF dilution.
   B. 6X PLX4720: make up stocks of 12 mM PLX4720 in DMSO, then dilute 1:1000 in media to make up 12 µM PLX4720 for use at 6X for the assay to avoid excessive DMSO toxicity.
   C. 6X PD184352: make up stocks of 6 mM PD184352 in DMSO, then dilute 1:1000 in media to make up 6 µM PD184352 for use at 6X for the assay to avoid excessive DMSO toxicity.
   D. DMSO dilution: dilute 1 µl DMSO with 999 µl media. Add 10 µl of this mix to DMSO dilution wells.
   A. These media dilutions are to avoid toxicity from excessive DMSO.

6. On day 7 after seeding, read GFP fluorescence and document bright-field and GFP images (BD, Pathway 435 Bioimager).
A. Subtract the average reading from media-only wells from the wells with cells.

7. Data analysis:

   A. Remove background fluorescence by subtracting the average reading from media-only wells from the wells with cells for each plate reading.
   B. Subtract the readings of day 1 from the other plates (day 4 and day 7) for the same wells.
   C. Average the quadruplicates.

D. Calculate the effect of PLX4720 and PD184352 in the presence or absence of HGF by normalizing the number of cells after 7 days of treatment (as measured by GFP fluorescence) to the number of cells present in the SK-MEL-5 vehicle control condition.

8. Repeat the experiment independently two additional times.

## Deliverables

- Data to be collected:

  1. Raw GFP fluorescence readings from days 1, 4, and 7.
  2. Normalized fluorescence proliferation data.
  3. Fluorescent and bright-field micrographs of cells from day 7.
  4. Bar chart of relative proliferation as a % of untreated control for all conditions. (Use data from Day 7 - Day 1 background) (Compare to Figure 2C)
  5. A semi-logarithmic graph of proliferation (log) vs time (linear) over 3 time points after seeding.

## Confirmatory analysis Plan

- Statistical Analysis:

  1. Compare the proliferation rate of PLX4720-treated cells treated with 0, 6.25, 12.5, 25, or 50 ng/ml HGF. Also compare each HGF cohort to the proliferation rate of vehicle-treated and untreated cells.

     A. One-way ANOVA

  2. Compare the proliferation rate of PD184352-treated cells treated with 0, 6.25, 12.5, 25, or 50 ng/ml HGF. Also compare each HGF cohort to the proliferation rate of vehicle-treated and untreated cells.

     A. One-way ANOVA

- Meta-analysis of original and replication attempt effect sizes:

  1. Compare the effect sizes of the original data to the replication data, using a meta-analytic approach to combine the original and replication effects which will be presented as a forest plot.

## Known differences from the original study

- The replication will only use one of the three melanoma cell lines used by the original authors, the SK-MEL-5 cell line. The replication will exclude SK-MEL-28 and G-361 cells.
- The replication will include an additional control, untreated SK-MEL-5 cells in addition to the vehicle (DMSO) treated SK-MEL-5 cells used in the original study.
- A Synergy HT Microplate Reader used instead of a Molecular Devices SpectraMax M5e Microplate Reader—both can detect GFP fluorescence and the Synergy HT Microplate Reader will be evaluated for range of detection (Protocol 1) and detection variability (Protocol 2)
- A BD Pathway 435 Bioimager used instead of a Zeiss Axio Observer.Z1—both are fluorescence microscopes with high-throughput screening capabilities.
- The replicating lab does not have a ViCell XR cell viability counter and thus will seed a larger number of cells per well (2800 instead of 2500 cells/well).

## Provisions for quality control

All data obtained from the experiment—raw data, data analysis, control data, and quality control data—will be made publicly available, either in the published manuscript or as an open access dataset available on the Open Science Framework (https://osf.io/p4lzc/).

- A lab from the Science Exchange network with extensive experience in conducting cell viability assays will perform these experiments.
- All cells will be sent for STR profiling to confirm identity and mycoplasma testing to confirm the lack of mycoplasma contamination.

- SK-MEL-5 cells will be confirmed to have at least 85% of the cells GFP-positive before the start of the experiment.

## Protocol 5: inhibitor proliferation assay

This experiment confirms that the rescue from drug sensitivity is due to HGF signaling by co-treating cells with crizotinib, an inhibitor of MET, the receptor tyrosine kinase for HGF, as seen in Figure 2D and Supplemental Figure 11.

### Sampling

- Run the experiment six times in total for a minimum power of 80%.

  1. See Power calculations section for details

- Each experiment has 10 cohorts:

  1. Each cohort consists of

     - SK-MEL-5 cells alone
     - SK-MEL-5 co-cultured with LL86 cells
     - SK-MEL-5 co-cultured with CCD-1090Sk cells

       - Each condition will be run in quadruplicate.

  2. The cohorts are treated with the following drugs:

     - Cohort 1: no drug treatment [additional control]
     - Cohort 2: treated with vehicle (DMSO) control
     - Cohort 3: treated with 0.2 µM crizotinib and vehicle
     - Cohort 4: treated with 0.2 µM PHA-665752 and vehicle [additional control]
     - Cohort 5: treated with 2 µM PLX4720 and vehicle
     - Cohort 6: treated with 2 µM PLX4720 and 0.2 µM crizotinib
     - Cohort 7: treated with 2 µM PLX4720 and 0.2 µM PHA-665752 [additional]
     - Cohort 8: treated with 1 µM PD184352 and vehicle
     - Cohort 9: treated with 1 µM PD184352 and 0.2 µM crizotinib
     - Cohort 10: treated with 1 µM PD184352 and 0.2 µM PHA-665752 [additional control]

### Materials and reagents

- Reagents that are different from ones originally used are noted with an asterisk (*).

| Reagent | Type | Manufacturer | Catalog # | Comments |
|---|---|---|---|---|
| pLEX-TRC206 SK-MEL-5 | Cells | Original authors | n/a | Engineered to express GFP |
| LL 86 cells | Cells | Original authors | n/a | Stromal cell line that secretes HGF |
| CCD-1090Sk cells | Cells | Original authors | n/a | Stromal cell line that does not secrete HGF |
| PLX4720 | Drug | Chemietek | CT-P4720 | BRAF inhibitor |
| PD184352 | Drug | Santa Cruz | sc-202759A | MEK inhibitor |
| crizotinib | Drug | Active Biochem | A-1031 | MET inhibitor |
| PHA-665752 | Drug | Sigma-Aldrich | PZ0147 | MET inhibitor [additional control] |
| 384-well clear-bottomed plates | Material | Corning | 3712 | |
| Phenol red free DMEM* | Medium | Sigma-Aldrich | D1145 | Original unspecified. This formulation of DMEM does not contain L-glutamine or sodium pyruvate, so these will be supplemented to the medium. |
| Sodium pyruvate solution* | Reagent | Sigma-Aldrich | S8636 | |
| FBS* | Reagent | Sigma-Aldrich | F4135 | Original unspecified |

*Table 5. Continued on next page*

*Table 5. Continued*

| Reagent | Type | Manufacturer | Catalog # | Comments |
|---------|------|-------------|-----------|----------|
| 100X Pen–Strep–Glut* | Reagent | Sigma-Aldrich | G1146 | Original from Invitrogen (15,140-122) |
| DMSO* | Reagent | Sigma-Aldrich | D8418 | Original unspecified |
| Synergy HT Microplate Reader* | Equipment | Bio-Tek | | Original equipment used: Molecular Devices SpectraMax M5e Microplate Reader |
| Biomek FX | Equipment | Beckman Coulter | | Communicated by authors. Original from Thermo Scientific (Combi reagent dispenser) and CyBio robotic liquid handler. |

## Procedure

1. On day 0, seed 40 wells of a 384-well clear-bottom plate with 1900 LL86 stromal cells in 20 µl phenol red free media, seed 40 wells with 1900 CCD-1090Sk stromal cells in 20 µl media, and seed 40 wells with phenol red free medium alone using an automated workstation.

Note:
   1. All cells will be sent for mycoplasma testing and STR profiling.
   2. Ensure at least 85% of SK-MEL-5 cells are GFP-positive before the start of the experiment. Cells can be enriched using FACS or antibiotics, however do not grow cells under antibiotic selection on a regular basis.
   3. Do not exceed a rate of 5–10 µl/s and do not let the tip end closer than 1 mm to the well bottom
   A. Total wells seeded: 120
   B. Fill wells with 60 µl/well of clear media in at least 2 rows and 2 columns around wells that are being included in the experiment.
   C. Medium of all cell lines for assay: phenol red free DMEM supplemented with 1 mM sodium pyruvate, 10% FBS, and 1X Pen–Strep–Glut.

2. In wells from Step 1, seed 1900 pLex-TRC206 SK-MEL-5 cells in 20 µl phenol red free medium per well using an automated workstation.
3. On day 1 after seeding, read GFP fluorescence (Synergy HT Microplate Reader).

   A. Subtract the average reading from media-only wells from the wells with cells.

4. Add appropriate drugs to each well (final volume = 60 µl).

   A. Formulation of drug stock solutions:

      i. 6X PLX4720: make up stocks of 12 mM PLX4720 in DMSO, then dilute 1:1000 in media to make up 12 µM PLX4720 for use at 6× for the assay to avoid excessive DMSO toxicity.
      ii. 6X PD184352: make up stocks of 6 mM PD184352 in DMSO, then dilute 1:1000 in media to make up 6 µM PD184352 for use at 6× for the assay to avoid excessive DMSO toxicity.
      iii. 6X crizotinib: make up stocks of 1.2 mM crizotinib in DMSO, then dilute 1:1000 in media to make up 1.2 µM PD184352 for use at 6× for the assay to avoid excessive DMSO toxicity.
      iv. 6X PHA-665752: make up stocks of 1.2 mM PHA-665752 in DMSO, then dilute 1:1000 in media to make up 1.2 µM PD184352 for use at 6× for the assay to avoid excessive DMSO toxicity.
      V. DMSO dilution: dilute DMSO 1:1000 in medium to avoid excessive DMSO toxicity.

   B. Cohort 1: add 20 µl phenol red free medium
   C. Cohort 2: add 10 µl DMSO dilution and 10 µl medium
   D. Cohort 3: add 10 µl 6X crizotinib and 10 µl medium
   E. Cohort 4: add 10 µl 6X PHA-665752 and 10 µl medium
   F. Cohort 5: add 10 µl 6X PLX4720 and 10 µl medium
   G. Cohort 6: add 10 µl 6X PLX4720 and 10 µl 6X crizotinib
   H. Cohort 7: add 10 µl 6X PLX4720 and 10 µl 6X PHA-665752
   I. Cohort 8: add 10 µl 6X PD184352 and 10 µl medium
   J. Cohort 9: add 10 µl 6X PD184352 and 10 µl 6X crizotinib
   K. Cohort 10: add 10 µl 6X PD184352 and 10 µl 6X PHA-665752

5. On day 4 after seeding, read GFP fluorescence.

   A. Subtract the average reading from media-only wells from the wells with cells.

6. Change the medium in relevant wells to 40 μl fresh media, then add appropriate drugs as per Step 4 using an automated workstation.

7. On day 7 after seeding, read GFP fluorescence and document bright-field and GFP images (BD, Pathway 435 Bioimager).

   A. Subtract the average reading from media-only wells from the wells with cells.

8. Data analysis:

   A. Remove background fluorescence by subtracting the average reading from media-only wells from the wells with cells for each plate reading.
   B. Subtract the readings of day 1 from the other plates (day 4 and day 7) for the same wells.
   C. Average the quadruplicates.
   D. Calculate the effect of PLX4720, PD184352, crizotinib, PHA-665752, PLX4720 + crizotinib, PLX4720 + PHA-665752, PD184352 + crizotinib, PD184352 + PHA-665752, DMSO, or untreated in the presence or absence of stromal cells by normalizing the number of cells after 7 days of treatment (as measured by GFP fluorescence) to the number of cells present in the vehicle control treated SK-MEL-5 cells alone condition.

9. Repeat experiment independently five additional times.

## Deliverables

- Data to be collected:

   1. Raw GFP fluorescence readings from days 1, 4, and 7.
   2. Normalized fluorescence proliferation data.
   3. Fluorescent and bright-field micrographs of cells from day 7.
   4. Bar chart of relative proliferation as a % of untreated control for all conditions. (Use data from Day 7 - Day 1 background) (compare to Figure F11)
   5. A. semi-logarithmic graph of proliferation (log) vs time (linear) over three time points after seeding.

## Confirmatory analysis plan

- Statistical analysis of replication data:

   1. Three-way ANOVA comparing the proliferation of vehicle-treated, PLX4720-treated, or PD184352-treated cells also treated with vehicle, crizotinib, or PHA-665752 cultured with or without stromal cells followed by:
   2. Two-way ANOVA comparing the proliferation of vehicle-treated cells treated with vehicle, crizotinib, or PHA-665752 cultured with or without stromal cells.
   3. Two-way ANOVA comparing the proliferation of PLX4720-treated cells treated with vehicle, crizotinib, or PHA-665752 cultured with or without stromal cells.

      1. Planned comparisons with the Bonferroni correction:

         - Vehicle-treated LL 86 cells compared to vehicle-treated no stromal cells
         - Vehicle-treated LL 86 cells compared to vehicle-treated CCD-1090Sk cells
         - Vehicle-treated LL 86 cells compared to crizotinib-treated LL 86 cells
         - Vehicle-treated LL 86 cells compared to PHA-665752-treated LL 86 cells

   4. Two-way ANOVA comparing the proliferation of PD184352-treated cells treated with vehicle, crizotinib, or PHA-665752 cultured with or without stromal cells.

      1. Planned comparisons with the Bonferroni correction:

         - Vehicle-treated LL 86 cells compared to crizotinib-treated LL 86 cells
         - Vehicle-treated LL 86 cells compared to PHA-665752-treated LL 86 cells

- Meta-analysis of original and replication attempt effect sizes:

    1. Compare the effect sizes of the original data to the replication data, using a meta-analytic approach to combine the original and replication effects which will be presented as a forest plot.

## Known differences from the original study

- Supplemental Figure 11 tests co-culture of SK-MEL-5 cells with 9 stromal cell lines. We have chosen LL86 cells, which showed the largest rescue of proliferation, and CCD-1090Sk cells, which showed the least rescue.
- Additional controls added by the replication team:

    1. Treatment with PHA-665752

        • In addition to inhibiting MET, crizotinib also targets ALK, ROS1, and RON. In order to confirm that the effects of crizotinib are due to targeting of MET, we will also use a more selective MET inhibitor, PHA-665752 (*Cui, 2014*; *Parikh et al., 2014*).

    2. The replication will include an additional control, untreated SK-MEL-5 cells in addition to the vehicle (DMSO) treated SK-MEL-5 cells used in the original study.

- A Synergy HT Microplate Reader used instead of a Molecular Devices SpectraMax M5e Microplate Reader

    1. Both can detect GFP fluorescence and the Synergy HT Microplate Reader will be evaluated for range of detection (Protocol 1) and detection variability (Protocol 2)

- A BD Pathway 435 Bioimager used instead of a Zeiss Axio Observer.Z1

    1. Both are fluorescence microscopes with high-throughput screening capabilities.

- The replicating lab does not have a ViCell XR cell viability counter and thus will seed a larger number of cells per well (1900 instead of 1700 cells/well).

## Provisions for quality control

All data obtained from the experiment - raw data, data analysis, control data, and quality control data - will be made publicly available, either in the published manuscript or as an open access dataset available on the Open Science Framework (https://osf.io/p4lzc/).

- A lab from the Science Exchange network with extensive experience in conducting cell viability assays will perform these experiments.
- All cells will be sent for STR profiling to confirm identity and mycoplasma testing to confirm the lack of mycoplasma contamination.
- SK-MEL-5 cells will be confirmed to have at least 85% of the cells GFP-positive before the start of the experiment.

## Protocol 6: inhibitor Western blot assay of ERK and AKT signaling

This experiment assesses the protein levels of various activated downstream pathway signaling component proteins in the presence or absence of HGF and drugs, as seen in Figure 4C and Supplemental Figure 19.

### Sampling

- Repeat the experiment six times in total for a minimum power of 85%.

    1. See Power calculations section for details

- Each experiment contains seven conditions:

    1. SK-MEL-5 cells treated with:

        • Untreated [additional control]
        • Vehicle (DMSO) control

- 2 µM PD184352
- 2 µM PLX4720
- 25 ng/ml HGF + vehicle
- 25 ng/ml HGF + 2 µM PD184352
- 25 ng/ml HGF + 2 µM PLX4720

## Materials and reagents
- Reagents that are different from ones originally used are noted with an asterisk (*).

| Reagent | Type | Manufacturer | Catalog # | Comments |
|---|---|---|---|---|
| Mouse anti-c-Met | Antibody | Cell Signaling | 3148 | 1:1000 dilution; 145 kDa |
| Rabbit anti-pMet Tyr 1349 | Antibody | Cell Signaling | 3133 | 1:1000 dilution; 145 kDa |
| Mouse anti-AKT | Antibody | Cell Signaling | 2920 | 1:2000 dilution; 60 kDa |
| Rabbit anti-pAKT | Antibody | Cell Signaling | 4060 | 1:2000 dilution; 60 kDa |
| Mouse anti-MEK | Antibody | Cell Signaling | 4694 | 1:1000 dilution; 45 kDa |
| Rabbit anti-pMEK | Antibody | Cell Signaling | 9154 | 1:1000 dilution; 45 kDa |
| Mouse anti-ERK | Antibody | Santa Cruz | 135900 | 1:200 dilution; 44,42 kDa |
| Rabbit anti-pERK | Antibody | Cell Signaling | 4370 | 1:2000 dilution; 44,42 kDa |
| Rabbit anti-GAPDH | Antibody | Cell Signaling | 2118 | 1:1000 dilution; 37 kDa Loading control |
| pLEX-TRC206 SK-MEL-5 | Cells | Original authors | n/a | Engineered to express GFP |
| PLX4720 | Drug | Chemietek | CT-P4720 | |
| PD184352 | Drug | Santa Cruz | sc-202759A | MET inhibitor |
| Odyssey Infrared Imaging System | Equipment | Li-COR | | |
| 6-well tissue culture plates* | Materials | Corning | 3516 | Original unspecified |
| DMEM* | Medium | Sigma-Aldrich | D6429 | Original from Invitrogen (10,569-010). |
| FBS* | Reagent | Sigma-Aldrich | F4135 | Original unspecified |
| 100X Pen–Strep* | Reagent | Sigma-Aldrich | P4333 | Original from Invitrogen (15,140-122) |
| DMSO* | Reagent | Sigma-Aldrich | D8418 | Original unspecified |
| HGF | Reagent | Sigma-Aldrich | H5791 | Original from RayBiotech (228-10,702-2) |
| PhosSTOP phosphatase inhibitor | Reagent | Roche | 04906837001 | |
| Complete mini protease inhibitor | Reagent | Roche | | |
| NuPAGE sample reducing agent | Reagent | Invitrogen | NP0009 | |
| TruPAGE 4–12% TEA-tricine gels* | Reagent | Sigma-Aldrich | PCG2003 | Original: NuPage (WG1402BOX) |
| TruPAGE TEA-Tricine SDS Running Buffer (20X) | Reagent | Sigma-Aldrich | PCG3001 | Original unspecified |
| TruPAGE LDS Sample Buffer (4X) | Reagent | Sigma-Aldrich | PCG3009 | Original unspecified |
| TruPAGE Transfer Buffer (20X) | Reagent | Sigma-Aldrich | PCG3011 | Original unspecified |
| Odyssey blocking buffer | Reagent | LI-COR | 927-40,000 | |
| Chameleon Kit Pre-stained Protein Ladder | Reagent | LI-COR | 928-90000 | Original unspecified |

*Table 6. Continued on next page*

*Table 6. Continued*

| Reagent | Type | Manufacturer | Catalog # | Comments |
|---|---|---|---|---|
| IRDye® 800CW Goat anti-RMouse IgG (H + L) | Antibody | Li-COR | 926-32210 | Original unspecified |
| IRDye 680RD Goat anti-Rabbit IgG (H + L) | Antibody | Li-COR | 926-68071 | Original unspecified |
| PBS, without MgCl2 and CaCl2 | Reagent | Sigma-Aldrich | D8537 | Original unspecified |
| IGEPAL CA-630 (NP-40 substitute) | Reagent | Sigma-Aldrich | I8896 | Original unspecified |
| Tween 20 | Reagent | Sigma-Aldrich | P1379 | Original unspecified |
| DC Protein Assay Kit II | Reagents | Bio-Rad | 500-0112 | |
| Odyssey Application Software | Software | Li-COR | | |
| Ponceau stain* | Reagent | Sigma-Aldrich | P3504 | Not included in the original study |
| Immobilon-FL PVDF membrane | Reagent | EMD Millipore | IPFL00010 | Original unspecified |

## Procedure

1. On day 0, plate $5 \times 10^5$ pLex-TRC206 SK-MEL-5 cells in 2 ml media per well for a total of 7 wells across 2 × 6-well plates.

Note:

   1. All cells will be sent for mycoplasma testing and STR profiling.
   2. Ensure at least 85% of SK-MEL-5 cells are GFP-positive before start of the experiment. Cells can be enriched using FACS or antibiotics, however do not grow cells under antibiotic selection on a regular basis.
   A. Medium of all cell lines for assay: DMEM supplemented with 10% FBS and 1× Pen–Strep.

2. On day 1 add the appropriate additives to each well.

   A. Formulation of stock solutions:

Note: these dilutions are to avoid toxicity from excessive DMSO.

   i. 1000X HGF: make a stock of 25 µg/ml HGF.
   ii. 1000X PLX4720: make a stock of 20 mM PLX4720 in DMSO, then dilute 1:10 in media to make a 2 mM PLX4720 working solution.
   iii. 1000X PD184352: make a stock of 20 mM PD184352 in DMSO, then dilute 1:10 in media to make a 2 mM working solution.
   iv. DMSO dilution: dilute DMSO 1:10 in medium.

   B. For media only: add 2 µl media
   C. For DMSO: add 2 µl DMSO dilution
   D. For 2 µM PD184352: add 2 µl 1000X PD184352
   E. For 2 µM PLX4720: add 2 µl 1000X PLX4720
   F. For 25 ng/ml HGF + DMSO: add 2 µl 1000X HGF and 2 µl DMSO dilution
   G. For 25 ng/ml HGF +2 µM PD184352: add 2 µl 1000X HGF and 2 µl 100X PD184352
   H. For 25 ng/ml HGF +2 µM PLX4720: add 2 µl 1000X HGF and 2 µl 100X PLX4720.

3. 24 hr after drug treatment, prepare cells for lysis.

   A. Quickly wash cells with ice-cold PBS and remove excess PBS.
   B. Add 0.5 ml or less of ice-cold lysis buffer to wells on ice.

   i. Lysis buffer: 50 mM Tris pH 7.4, 150 mM NaCl, 2 mM EDTA, 1% NP-40, 1 mg/ml NaF, and one pellet per 10 ml each of PhosSTOP phosphatase inhibitor and complete mini protease inhibitor.

   C. Scrape cells off dish with cell scraper.
   D. Collect cells in a 1.5 ml centrifuge tube on ice.

 E. Incubate on ice for 30 min with periodic vortexing.
 F. Spin down at 4°C and remove supernatant into separate tube.

4. Determine protein concentration by using the DC Protein Assay Kit II following manufacturer's instructions.
5. Mix 50 µg total cell lysate with NuPAGE sample reducing agent and run on two 4–12% TEA-tricine gels with a protein molecular weight marker at 120 V.
6. Transfer onto membrane using replicating lab's transfer protocol.
7. After the transfer, stain the membrane with Ponceau to visualize the transferred protein. Image membrane, then wash out the Ponceau stain [additional quality control step].
8. Wet membrane with PBS for 5 min, then block membranes in Odyssey blocking buffer (LI-COR, 927-40,000) following manufacturer's instructions.
9. Probe membrane with the following primary antibodies diluted in Odyssey blocking buffer at 4°C with gentle shaking, overnight.

 A. Mouse anti-c-Met (Cell Signaling, 3148); 1:1000; 145 kDa
 B. Rabbit anti-pMet Tyr 1349 (Cell Signaling, 3133); 1:1000; 145 kDa
 C. Mouse anti-AKT (Cell Signaling, 2920); 1:2000; 60 kDa
 D. Rabbit anti-pAKT (Cell Signaling, 4060); 1:2000; 60 kDa
 E. Mouse anti-MEK (Cell Signaling, 4694); 1:1000; 45 kDa
 F. Rabbit anti-pMEK (Cell Signaling, 9154); 1:1000; 45 kDa
 G. Mouse anti-ERK (Santa Cruz, 135900); 1:200; 44,42 kDa
 H. Rabbit anti-pERK (Cell Signaling, 4370); 1:2000; 44,42 kDa
 I. Rabbit anti-GAPDH (Cell Signaling, 2118); 1:1000; 37 kDa

 i. Loading control

 J. Note: multiple gels will need to be run to probe for this many proteins. Do not strip between probing with different phospho antibodies, just wash membrane well (4 × 10 min PBS-T) and then add next antibody. Suggest grouping as follows:

 i. Gel 1: Probe pAKT [rabbit 60 kDa], then pMEK [rabbit 45 kDa], then AKT [mouse 60 kDa], then MEK [mouse 45 kDa], then GAPDH [rabbit 37 kDa].
 ii. Gel 2: Probe pMet Tyr 1349 [rabbit 145 kDa], then pERK [rabbit 44,42 kDa], then c-Met [mouse 145 kDa], then ERK [mouse 44,42 kDa], then GAPDH [rabbit 37 kDa].

10. Wash membranes in PBS +0.1% Tween 20 4 × 5 min.
11. Detect primary antibodies with anti-rabbit or anti-mouse IRDye secondary antibodies (LICOR) diluted in Odyssey blocking buffer for 30–60 min protected from light following manufacturer's instructions.
12. Wash membranes in PBS +0.1% Tween 20 4 × 5 min.
13. Rinse membrane with PBS to remove residual Tween 20.
14. Detect near infrared fluorescence with the Odyssey Infrared Imaging System.
15. Quantify signal intensity with Odyssey Application Software.

 A. For each antibody subtract background intensity from values and then divide by the GAPDH loading control.
 B. Calculate the effect of PLX4720, PD184352, or vehicle in the presence or absence of HGF by normalizing the band intensities (after background and loading correction) to the band intensity of the SK-MEL-5 vehicle control condition.

16. Repeat experiment independently five additional times.

## Deliverables

- Data to be collected:

 1. Odyssey images of probed membranes (full images with ladder).
 2. Raw and quantified signal intensities normalized for GAPDH loading and total pan-protein levels.
 3. Bar graphs of normalized mean signal intensities (compare to Figure 19).

## Confirmatory analysis plan

- Statistical analysis of replication data:

  1. Two-way ANOVA comparing the relative phopho-AKT band intensities of cells treated with vehicle, PLX4720, or PD184352 in the presence or absence of HGF.

     A. Planned comparisons with the Bonferroni correction:

     - PLX4720-treated cells in the absence of HGF compared to PLX4720-treated cells in the presence of HGF.

  2. Two-way ANOVA comparing the relative phopho-ERK band intensities of cells treated with vehicle, PLX4720, or PD184352 in the presence or absence of HGF.

     A. Planned comparisons with the Bonferroni correction:

     - PLX4720-treated cells in the absence of HGF compared to PLX4720-treated cells in the presence of HGF.

  3. Two-way ANOVA comparing the relative phopho-MET (Tyr1349) band intensities of cells treated with vehicle, PLX4720, or PD184352 in the presence or absence of HGF.

     A. Planned comparisons with the Bonferroni correction:

     - Cells treated in the absence of HGF and treated with vehicle, PLX4720, or PD184352 compared to cells treated in the presence of HGF and treated with vehicle, PLX4720, or PD184352.

- Meta-analysis of original and replication attempt effect sizes:

  1. Compare the effect sizes of the original data to the replication data and use a meta-analytic approach to combine the original and replication effects, which will be presented as a forest plot.

## Known differences from the original study

- Provider lab transfer protocol used instead of iBlot Gel Transfer Device (Invitrogen, IB1001) using Program 4—both are capable of transferring protein efficiently, and to determine completeness of the transfer the gel may be stained (Step 8).
- The replication will include an additional control, untreated SK-MEL-5 cells in addition to the vehicle (DMSO) treated SK-MEL-5 cells used in the original study.
- The replication will not include the pMet Tyr1234/5, RAF1, and pRAF1 antibodies included in the original study.

## Provisions for quality control

All data obtained from the experiment—raw data, data analysis, control data, and quality control data—will be made publicly available, either in the published manuscript or as an open access dataset available on the Open Science Framework (https://osf.io/p4lzc/).

- A lab from the Science Exchange network with extensive experience in conducting cell viability assays and performing Western blots will perform these experiments.
- All cells will be sent for STR profiling to confirm identity and mycoplasma testing to confirm the lack of mycoplasma contamination.
- SK-MEL-5 cells will be confirmed to have at least 85% of the cells GFP-positive before the start of the experiment.

## Power calculations

All calculations are determined in order to reach at least 80% power.

### Protocol 1

No power calculations required.

## Protocol 2

No power calculations required.

## Protocol 3

Summary of original data:

Note: original data values were shared by authors.

| SK-MEL-5 cells only | Mean | SEM | SD | N |
|---|---|---|---|---|
| Unconditioned medium | 30.7 | 6.06 | 10.5 | 3 |
| CCD-1090Sk conditioned medium | 32.0 | 0.73 | 1.26 | 3 |
| PC60163A1 conditioned medium | 83.3 | 11.66 | 20.3 | 3 |
| LL 86 conditioned medium | 83.6 | 7.26 | 12.6 | 3 |

- Standard deviation was calculated using the formula, SD = SEM*(SQRT n)

## Test family

- ANOVA: fixed effects, omnibus, one-way, alpha error = 0.05
  - Power calculations were performed from effects reported in the original study using G*Power software (version 3.1.7) (*Faul et al., 2007*).
  - ANOVA F statistic calculated with Graphpad Prism 6.0
  - Partial $\eta^2$ calculated from Lakens (2013)

## Power calculations for replication

| Groups | F test statistic | Partial $\eta^2$ | Effect size $f$ | A priori power | Total sample size |
|---|---|---|---|---|---|
| Unconditioned, CCD-1090Sk, PC60163A1, and LL 86 conditioned medium | $F(3,8) = 15.9095$ | 0.8564 | 2.442087 | 93.7%[a] | 8[a] (4 groups) |

[a]A total sample size of 16 will be used based on the planned comparison calculations making the power 99.9%.

## Test family

- Two tailed *t*-test; difference between two independent means Bonferroni's correction: alpha error = 0.0125.
  - Calculations were performed from effects reported in the original study using G*Power software (version 3.1.7) (*Faul et al., 2007*).

## Power calculations for replication

| Group 1 | Group 2 | Effect size $d$ | A Priori power | Group 1 sample size | Group 2 sample size |
|---|---|---|---|---|---|
| Unconditioned medium | PC60163A1 conditioned medium | 3.254798 | 80.9% | 4 | 4 |
| Unconditioned medium | LL 86 conditioned medium | 4.561277 | 80.7%[a] | 3[a] | 3[a] |
| CCD-1090Sk conditioned medium | PC60163A1 conditioned medium | 3.566986 | 88.0% | 4 | 4 |
| CCD-1090Sk conditioned medium | LL 86 conditioned medium | 5.762799 | 94.8%[b] | 3[b] | 3[b] |

[a]4 per group will be used based on the other comparisons making the power 98.3%.
[b]4 per group will be used based on the other comparisons making the power 99.9%.

## Protocol 4

### Summary of original data

Note: original data values were shared by authors.

| PLX4720-treated SK-MEL-5 cells | Mean | SEM | SD | N |
|---|---|---|---|---|
| 0 ng/ml HGF | 32.1 | 11.0 | 19.1 | 3 |
| 6.25 ng/ml HGF | 73.5 | 3.09 | 5.35 | 3 |
| 12.5 ng/ml HGF | 84.3 | 7.27 | 12.6 | 3 |
| 25 ng/ml HGF | 93.7 | 13.0 | 22.5 | 3 |
| 50 ng/ml HGF | 96.9 | 8.56 | 14.8 | 3 |

- Standard deviation was calculated using the formula, $SD = SEM*(SQRT\ n)$

### Test family

- ANOVA: fixed effects, omnibus, one-way, alpha error = 0.05
  - Power calculations were performed from effects reported in the original study using G*Power software (version 3.1.7) (*Faul et al., 2007*).
  - ANOVA F statistic calculated with Graphpad Prism 6.0
  - Partial $\eta^2$ calculated from Lakens (2013)

### Power calculations for replication

| Groups | F test statistic | Partial $\eta^2$ | Effect size $f$ | A priori power | Total sample size |
|---|---|---|---|---|---|
| 0, 6.25, 12.5, 25, and 50 ng/ml HGF | $F(4,10) = 8.0796$ | 0.7637 | 1.797751 | 97.7%[a] | 10[a] (5 groups) |

[a]A total sample size of 15 will be used as a minimum making the power 99.9%.

### Summary of original data

Note: original data values were shared by authors.

| PD184352-treated SK-MEL-5 cells | Mean | SEM | SD | N |
|---|---|---|---|---|
| 0 ng/ml HGF | 30.3 | 9.79 | 17 | 3 |
| 6.25 ng/ml HGF | 58.1 | 12.7 | 22.0 | 3 |
| 12.5 ng/ml HGF | 65.8 | 4.52 | 7.83 | 3 |
| 25 ng/ml HGF | 80.1 | 0.66 | 1.14 | 3 |
| 50 ng/ml HGF | 89.7 | 3.19 | 5.53 | 3 |

- Standard deviation was calculated using the formula, $SD = SEM*(SQRT\ n)$

### Test family

- ANOVA: fixed effects, omnibus, one-way, alpha error = 0.05
  - Power calculations were performed from effects reported in the original study using G*Power software (version 3.1.7) (*Faul et al., 2007*).
  - ANOVA F statistic calculated with Graphpad Prism 6.0
  - Partial $\eta^2$ calculated from Lakens (2013)

## Power calculations for replication

| Groups | F test statistic | Partial η² | Effect size f | A priori power | Total sample size |
|---|---|---|---|---|---|
| 0, 6.25, 12.5, 25, and 50 ng/ml HGF | $F(4,10) = 9.0493$ | 0.7835 | 1.902351 | 86.8%[a] | 10[a] (5 groups) |

[a]A total sample size of 15 will be used as a minimum making the power 99.9%.

## Protocol 5
Summary of original data

Note: numbers were shared by original authors.

| Stromal cells | BRAF/MEK inhibitor | MET inhibitor | Mean | SEM | SD | N |
|---|---|---|---|---|---|---|
| None | Vehicle | Vehicle | 100 | 0.00 | 0.00 | 3 |
| None | Vehicle | Crizotinib | 97.4 | 0.39 | 0.70 | 3 |
| None | Vehicle | PHA-665752 | 97.4[a] | 0.39[a] | 0.70[a] | 3[a] |
| None | PLX4720 | Vehicle | 32.2 | 10.8 | 18.7 | 3 |
| None | PLX4720 | Crizotinib | 28.4 | 8.81 | 15.3 | 3 |
| None | PLX4720 | PHA-665752 | 28.4[a] | 8.81[a] | 15.3[a] | 3[a] |
| None | PD184352 | Vehicle | 24.4 | 13.2 | 22.9 | 3 |
| None | PD184352 | Crizotinib | 26.6 | 2.73 | 4.70 | 3 |
| None | PD184352 | PHA-665752 | 26.6[a] | 2.73[a] | 4.70[a] | 3[a] |
| LL 86 | Vehicle | Vehicle | 99.2 | 2.05 | 3.60 | 3 |
| LL 86 | Vehicle | Crizotinib | 99.1 | 3.63 | 6.30 | 3 |
| LL 86 | Vehicle | PHA-665752 | 99.1[a] | 3.63[a] | 6.30[a] | 3[a] |
| LL 86 | PLX4720 | Vehicle | 91.0 | 9.32 | 16.1 | 3 |
| LL 86 | PLX4720 | Crizotinib | 33.4 | 7.28 | 12.6 | 3 |
| LL 86 | PLX4720 | PHA-665752 | 33.4[a] | 7.28[a] | 12.6[a] | 3[a] |
| LL 86 | PD184352 | Vehicle | 56.9 | 11.1 | 19.2 | 3 |
| LL 86 | PD184352 | Crizotinib | 25.4 | 3.64 | 6.30 | 3 |
| LL 86 | PD184352 | PHA-665752 | 25.4[a] | 3.64[a] | 6.30[a] | 3[a] |
| CCD-1090Sk | Vehicle | Vehicle | 99.7 | 0.80 | 1.40 | 3 |
| CCD-1090Sk | Vehicle | Crizotinib | 100.8 | 3.40 | 5.90 | 3 |
| CCD-1090Sk | Vehicle | PHA-665752 | 100.8[a] | 3.40[a] | 5.90[a] | 3[a] |
| CCD-1090Sk | PLX4720 | Vehicle | 31.1 | 8.40 | 14.5 | 3 |
| CCD-1090Sk | PLX4720 | Crizotinib | 27.1 | 6.10 | 10.6 | 3 |
| CCD-1090Sk | PLX4720 | PHA-665752 | 27.1[a] | 6.10[a] | 10.6[a] | 3[a] |
| CCD-1090Sk | PD184352 | Vehicle | 23.7 | 10.2 | 17.7 | 3 |
| CCD-1090Sk | PD184352 | Crizotinib | 26.9 | 10.1 | 17.5 | 3 |
| CCD-1090Sk | PD184352 | PHA-665752 | 26.9[a] | 10.1[a] | 17.5[a] | 3[a] |

[a]All PHA-665752 treatment values were made the same as the corresponding crizotinib treatment as these inhibitors are assumed to have the same effect. PHA-665752 is an additional MET inhibitor added to the experimental design.

- Standard deviation was calculated using the formula, SD = SEM*(SQRT n)

## Test family

- 3-way ANOVA between subjects: fixed effects, special, main effects and interactions, alpha error = 0.05

- Power calculations were performed from effects reported in the original study using G*Power software (version 3.1.7) (*Faul et al., 2007*).
- ANOVA F statistic calculated with R software 3.1.1 (R Core Team, 2014)
- Partial $\eta^2$ calculated from Lakens (2013)

## Power calculations for replication

| Groups | F test statistic | Partial $\eta^2$ | Effect size $f$ | A priori power | Total sample size |
|---|---|---|---|---|---|
| All 27 groups | F(8,54) = 3.6903 (interaction)[b] | 0.3535 | 0.739453 | 80.4%[a] | 43[a] (27 groups) |

[a]A total sample size of 162 will be used based on the planned comparison calculations making the power 99.9%.
[b]10,000 simulations were run using the summary data to randomly assign data values and the interaction F statistic was computed for a 3-way ANOVA between subjects design. The average F statistic was calculated and used in the power calculations.

## Test family

- 2-way ANOVA between subjects: fixed effects, special, main effects, and interactions, alpha error = 0.05

  - Power calculations were performed from effects reported in the original study using G*Power software (version 3.1.7) (*Faul et al., 2007*).
  - ANOVA F statistic calculated with Graphpad Prism 6.0
  - Partial $\eta^2$ calculated from Lakens (2013)

## Power calculations for replication (BRAF/MEK inhibitor)

| Groups | F Test statistic | Partial $\eta^2$ | Effect size $f$ | A Priori power | Total sample size |
|---|---|---|---|---|---|
| All 9 vehicle groups | F(4,18) = 0.9381 (interaction) | 0.1725 | 0.495450[a] | 80.0%[a] | 54[a] (9 groups) |
| All 9 vehicle groups | F(2,18) = 0.5678 (main effect: stromal cells) | 0.0593 | 0.436865[a] | 80.0%[a] | 54[a] (9 groups) |
| All 9 vehicle groups | F(2,18) = 0.9546 (main effect: MET inhibitor) | 0.0959 | 0.436865[a] | 80.0%[a] | 54[a] (9 groups) |
| All 9 PLX4720 groups | F(4,18) = 4.7285 (interaction) | 0.5124 | 1.025115 | 82.1%[b] | 19[b] (9 groups) |
| All 9 PD184352 groups | F(4,18) = 1.8076 (interaction) | 0.2866 | 0.633828 | 80.8%[c] | 36[c] (9 groups) |

[a]A sensitivity calculation was performed since the original data showed a non-significant effect with the computed effect size shown that can be detected with 80% power.
[b]54 total will be used based on the planned comparison calculations making the power 99.9%.
[c]54 total will be used based on the planned comparison calculations making the power 96.0%.

## Test family

- Two tailed *t*-test; difference between two independent means, Bonferroni's correction: alpha error = 0.0125.

  - Power calculations were performed for effects reported in the original study using G*Power software (version 3.1.7) (*Faul et al., 2007*).

## Power calculations for replication (PLX4720 group)

| Group 1 | Group 2 | Effect size *d* | A Priori power | Group 1 sample size | Group 2 sample size |
|---|---|---|---|---|---|
| No stromal cells treated with PLX4720 and vehicle | LL 86 stromal cells treated with PLX4720 and vehicle | 3.369918 | 83.8%[a] | 4[a] | 4[a] |
| LL 86 stromal cells treated with PLX4720 and vehicle | LL 86 stromal cells treated with PLX4720 and crizotinib | 3.984418 | 94.2%[b] | 4[b] | 4[b] |
| LL 86 stromal cells treated with PLX4720 and vehicle | CCD-1090Sk stromal cells treated with PLX4720 and vehicle | 3.909692 | 93.4%[c] | 4[c] | 4[c] |
| LL 86 stromal cells treated with PLX4720 and vehicle | LL 86 stromal cells treated with PLX4720 and PHA-665752 | 3.984418 | 94.2%[d] | 4[d] | 4[d] |

[a]6 per group will be used based on the PD184352 planned comparisons making the power 99.1%.
[b]6 per group will be used based on the PD184352 planned comparisons making the power 99.9%.
[c]6 per group will be used based on the PD184352 planned comparisons making the power 99.9%.
[d]6 per group will be used based on the PD184352 planned comparisons making the power 99.9%.

## Test family

- Two tailed *t*-test; difference between two independent means, Bonferroni's correction: alpha error = 0.025.

  - Power calculations were performed for effects reported in the original study using G*Power software (version 3.1.7) (*Faul et al., 2007*).

## Power calculations for replication (PD184352 group)

| Group 1 | Group 2 | Effect size *d* | A Priori power | Group 1 sample size | Group 2 sample size |
|---|---|---|---|---|---|
| LL 86 stromal cells treated with PD184352 and vehicle | LL 86 stromal cells treated with PD184352 and crizotinib | 2.204550 | 86.0% | 6 | 6 |
| LL 86 stromal cells treated with PD184352 and vehicle | LL 86 stromal cells treated with PD184352 and PHA-665752 | 2.204550 | 86.0% | 6 | 6 |

## Protocol 6
Summary of original data

Note: numbers were estimated from bar chart in Supplemental Figure S19.

### pAKT

| Growth Factor | BRAF/MEK inhibitor | Mean | SEM | SD | N |
|---|---|---|---|---|---|
| Vehicle | Vehicle | 1.00 | 0.00 | 0.00 | 3 |
| Vehicle | PD184352 | 0.84 | 0.33 | 0.57 | 3 |
| Vehicle | PLX4720 | 1.22 | 0.58 | 1.00 | 3 |
| HGF | Vehicle | 5.11 | 0.56 | 0.97 | 3 |
| HGF | PD184352 | 11.56 | 5.22 | 9.04 | 3 |
| HGF | PLX4720 | 9.11 | 2.11 | 3.65 | 3 |

## pERK

| Growth Factor | BRAF/MEK inhibitor | Mean | SEM | SD | N |
|---|---|---|---|---|---|
| Vehicle | Vehicle | 1.00 | 0.00 | 0.00 | 3 |
| Vehicle | PD184352 | 0.00 | 0.00 | 0.00 | 3 |
| Vehicle | PLX4720 | 0.08 | 0.07 | 0.12 | 3 |
| HGF | Vehicle | 1.60 | 0.12 | 0.21 | 3 |
| HGF | PD184352 | 0.39 | 0.21 | 0.36 | 3 |
| HGF | PLX4720 | 1.61 | 0.65 | 1.13 | 3 |

## pMET(Tyr1349)

| Growth Factor | BRAF/MEK inhibitor | Mean | SEM | SD | N |
|---|---|---|---|---|---|
| Vehicle | Vehicle | 1.00 | 0.00 | 0.00 | 3 |
| Vehicle | PD184352 | 2.91 | 2.00 | 3.46 | 3 |
| Vehicle | PLX4720 | 2.87 | 2.91 | 5.04 | 3 |
| HGF | Vehicle | 9.44 | 5.11 | 8.85 | 3 |
| HGF | PD184352 | 16.44 | 6.58 | 11.40 | 3 |
| HGF | PLX4720 | 13.73 | 5.67 | 9.82 | 3 |

- Standard deviation was calculated using the formula, SD = SEM*(SQRT 3)]

## Test family

- 2-way ANOVA between subjects: fixed effects, special, main effects, and interactions, alpha error = 0.05

  - Power calculations were performed from effects reported in the original study using G*Power software (version 3.1.7) (*Faul et al., 2007*).
  - ANOVA F statistic calculated with Graphpad Prism 6.0
  - Partial $\eta^2$ calculated from Lakens (2013)

## Power calculations for replication

| Groups | F Test statistic | Partial $\eta^2$ | Effect size $f$ | A Priori power | Total sample size |
|---|---|---|---|---|---|
| pAKT | F(1,12) = 15.9141 (main effect: growth factor) | 0.5701 | 1.151574 | 85.9%[a] | 11[a] (6 groups) |
| pERK | F(1,12) = 13.0042 (main effect: growth factor) | 0.5201 | 1.041042 | 85.0%[b] | 12[b] (6 groups) |
| pERK | F(2,12) = 7.5790 (main effect: BRAF/MEK inhibitor) | 0.5581 | 1.123813 | 82.9%[c] | 13[c] (6 groups) |
| pMET (Tyr1349) | F(1,12) = 9.4520 (main effect: growth factor) | 0.4406 | 0.887485 | 82.8%[d] | 14[d] (6 groups) |

[a]36 total will be used based on the planned comparison calculations making the power 99.9%.
[b]36 total will be used based on the planned comparison calculations making the power 99.9%.
[c]36 total will be used based on the planned comparison calculations making the power 99.9%.
[d]36 total will be used based on the planned comparison calculations making the power 99.9%.

## Test family

- Two tailed *t*-test; difference between two independent means, Bonferroni's correction: alpha error = 0.05.

o Note: calculations were performed for effects reported in the original study using G*Power software (version 3.1.7) (*Faul et al., 2007*).

## Power calculations for replication (pAKT group)

| Group 1 | Group 2 | Effect size $d$ | A priori power | Group 1 sample size | Group 2 sample size |
|---|---|---|---|---|---|
| pAKT, vehicle, PLX4720 | pAKT, HGF, PLX4720 | 2.943958 | 93.1%[a] | 4[a] | 4[a] |

[a]6 per group will be used based on the pERK planned comparisons making the power 99.5%.
Note: HGF/PD184352 compared to vehicle/PD184352 is not included as the number of needed samples is too large.

### Test family

- Two tailed *t*-test; difference between two independent means, Bonferroni's correction: alpha error = 0.05.

  o Note: calculations were performed for effects reported in the original study using G*Power software (version 3.1.7) (*Faul et al., 2007*).

## Power calculations for replication (pERK group)

| Group 1 | Group 2 | Effect size $d$ | A priori power | Group 1 sample size | Group 2 sample size |
|---|---|---|---|---|---|
| pERK, vehicle, PLX4720 | pERK, HGF, PLX4720 | 1.910859 | 84.6% | 6 | 6 |

Note: HGF/PD184352 compared to vehicle/PD184352 is not included as the number of needed samples is too large.

### Test family

- Two tailed *t*-test; difference between two independent means, Bonferroni's correction: alpha error = 0.05.

  o Note: calculations were performed for effects reported in the original study using G*Power software (version 3.1.7) (*Faul et al., 2007*).

## Power calculations for replication (pMET(Tyr1349) group)

| Group 1 | Group 2 | Effect size $d$ | A priori power | Group 1 sample size | Group 2 sample size |
|---|---|---|---|---|---|
| All 3 vehicle (no HGF) conditions | All 3 HGF conditions | 1.581545 | 83.7%[a] | 8[a] (3 conditions) | 8[a] (3 conditions) |

[a]18 per group (6/condition) will be used based on the pERK planned comparisons making the power 99.6%.

## Acknowledgements

The Reproducibility Project: Cancer Biology core team would like to thank the original authors, in particular Ravid Straussman and Michal Barzily-Rokni, for generously sharing critical information and reagents to ensure the fidelity and quality of this replication attempt. We would also like to thank the following companies for generously donating reagents to the Reproducbility Project: Cancer Biology: American Type Culture Collection (ATCC), BioLegend, Charles River Laboratories, Corning Incorporated, DDC Medical, EMD Millipore, Harlan Laboratories, LI-COR Biosciences, Mirus Bio, Novus Biologicals, and Sigma–Aldrich.

## Additional information

### Group author details

**Reproducibility Project: Cancer Biology**

Elizabeth Iorns: Science Exchange, Palo Alto, United States; William Gunn: Mendeley, London, United Kingdom; Fraser Tan: Science Exchange, Palo Alto, United States; Joelle Lomax: Science Exchange, Palo Alto, United States; Timothy Errington: Center for Open Science, Charlottesville, United States

### Competing interests

DB: This is a Science Exchange associated lab. RP:CB: EI, FT and JL are employed by and hold shares in Science Exchange Inc. The other authors declare that no competing interests exist.

### Funding

| Funder | Author |
| --- | --- |
| Laura and John Arnold Foundation | Reproducibility Project: Cancer Biology |

The Reproducibility Project: Cancer Biology is funded by the Laura and John Arnold Foundation, provided to the Center for Open Science in collaboration with Science Exchange. The funder had no role in study design or the decision to submit the work for publication.

### Author contributions

DB, SLB, Drafting or revising the article; RP:CB, Conception and design, Drafting or revising the article

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
