## [Decision Letter]

Thank you for sending your work entitled “Registered report: Tumour
micro-environment elicits innate resistance to RAF inhibitors through HGF
secretion” for consideration at *eLife*. Your Registered report
has been reviewed by Charles Sawyers, Ravid Straussman as one of the original authors,
and a biostatistician.

Charles Sawyers has assembled the following comments to help you prepare a revised
submission.

All the reviewers agree that you have appropriately identified the most salient features
of Straussman et al. for replication and that the replication experiments are well
designed. Protocols 3, 4 and 5 are the key experiments (stromal conditioned media
rescue, recombinant HGF rescue and inhibition of rescue with crizotinib). One reviewer
felt that protocol 6 (survey of other signaling pathways activated by HGF) was
optional.

1) Two reviewers felt that more attention should be given to the [5] publication that reportedly failed to confirm
a correlation between HGF expression and outcome (Figure 3 in Straussman et al.).

Specifically:

a. It should be noted in the text that the same group did replicate some other key
findings of Straussman et al. – the presence of HGF in human melanoma tumors (in
both melanoma cells and stromal cells) and the finding that HGF is significantly
enhanced in disease progression.

b. The testing for a correlation between pre-treatment HGF and clinical outcome was done
by Lezcano et al. using a cohort of 23 pre-treatment samples. While we fully support the
claim by Lezcano et al. that “rigorous validation studies are thus indicated for
approaches that seek to personalize such therapies to maximize therapeutic
efficacy,” we wonder if testing of 23 samples can be considered as rigorous. As
no power calculations are mentioned in Lezcano et al., we would like to see some
discussion of whether Lezcano et al. were sufficiently powered to make positive or
negative associations. If not, how large would the sample sizes need to be?

c. Wilson et al. (PMID: 22763448) tested the correlation between plasma HGF and PFS/OS
on 126 melanoma patients and did find a statistically significant negative correlation
that supports the findings in Straussman et al. As this is the only available big cohort
testing HGF and clinical outcome on BRAFi, this should be adding it to the literature
summary in the introduction.

2) We are aware of 2 groups that have directly replicated several of the in vitro
experiments of the paper and have published some results. These should be added these to
the literature summary.

a. A group from Amgen attempted to directly replicate the key findings from Straussman
et al. Their findings can be found here: http://cancerres.aacrjournals.org/cgi/content/meeting_abstract/73/8_MeetingAbstracts/3405.
They show that HGF can rescue melanoma cell lines from BRAFi and MEKi and that this
rescue is attenuated by METi.

b. A group from the University of Illinois was able to demonstrate that c-MET inhibition
is synergistic with BRAF inhibition in melanoma cell lines: http://cancerres.aacrjournals.org/cgi/content/meeting_abstract/73/8_MeetingAbstracts/2078

3) Regarding statistical power, we also have the following suggestion:

While it is very useful for you to leverage the previously reported effects to compute
minimum power a priori, what you really need is to guarantee a minimum power on your own
data. This can be done, a priori, by including some cross-study variation. This will be
helpful for you to plan on the number of replicates and so forth. Papers by Giovanni
Parmigiani and collaborators at the Dana Farber provide some estimates about cross-study
variation that could be used for this purpose. Worst case, you should budget some
additional variability because of cross-study reproducibility, and increase the sample
size as appropriate. We also want you to compute and report power post-hoc/on-the-fly on
your own data. Some minimum power should be guaranteed using summaries of your own
data.

Comments on the specific protocols:

Protocols 1 and 2 - We think that the protocols are mixed and experiment detailed in
protocol 2 should be protocol 1 and vice versa. This should be corrected. Below we refer
to the protocols as they appear in the file that we received.

Protocol 1

• When growing SK-MEL-5-GFP cells make sure that >85% of cells are GFP
labeled. If number of GFP positive cells are dropping one can use FACS or antibiotics to
enrich again for GFP positive cells. We did not grow the cells under antibiotic
selection on a regular basis.

• Microplate reader used is different from original and should be labeled with a
*.

• We used Corning #3712 plates and did specify that in the methods section.
Please remove the * and remove the comment: “Original
unspecified”.

• 1c – as specified in the methods section we maintained cells in DMEM
from Invitrogen (#10569-010). While using phenol-red free DMEM for the screens is a
good idea (we did the same) I would recommend supplementing it with sodium pyruvate as
the DMEM that we used had Sodium pyruvate in it. When using Phenol-red free media we
used to add Sodium pyruvate from Cellgro (Cat #25-000-CI) to a final of 1 mM.

• 1d - we have plated cells on 384-well plates using the Combi cell platter
(http://www.thermo.com.cn/Resources/201306/21143420640.pdf). This resulted
in very accurate plating. I don't know how the replicating lab is planning to plate
cells on 384-well plates. If manual plating is planned make sure that no air bubble is
present at the bottom of the well after plating as this can frequently occur for those
unexperienced with manual plating of 384-well plates.

Protocol 2

• Read GFP only after cells have completely settled down. As indicated in the
paper we used to plate cells on day 0 and read GFP for the first time on day 1.

• Read GFP from wells with media and no cells as well. Before analyzing results
subtract reading from clear–media wells from wells that have cells. We noticed
that reading from media-only wells can change from day from day and thus subtract the
reading from media-only wells from wells with cells. To this end we always make sure to
have media-only wells on each plate with a total volume that is equivalent to
test-wells. This remark is true to all experiments in all protocols.

Protocol 3

• 2b - Seems like 50ul and not 60ul is a better control for the wells that will
have 20ul of cancer cells +20ul of PCM +10ul of drug.

• From the protocol it seems like stromal cells are plated once. I have plated
stromal cells 3 times (each time 3 days before I needed it) to make sure that I have
fresh PCM on days 0,1 & 4.

• This protocol involves a few cycles of media change in 384-well plates. We have
done so using a CyBi robotic liquid handler. Do the replicating lab plan to use a
robotic liquid handler? From my experience it is not easy to take out the exact same
amount of media from 384-wells manually making sure not to touch the bottom and disturb
the cells. If a robotic plate handler can be used I would recommend using it, as manual
handling of hundreds of 384-wells might be a source of a lot of noise in the experiment.
Lastly - both extraction and addition of liquid from the wells should be done gently.
Cells are under the treatment of BRAFi and might be displaced more easily that
non-treated cells. If using a robotic system please do not exceed a rate of
5-10μl/s and do not let the tip end closer than 1 mm to the well bottom.

• 8a - Subtract the reading from media only wells first and only then subtract
reading of day 1 from day 7. This remark is true for all protocols.

Protocol 4

• 1a – please correct number to match your planned 2500 cells/well.

• 5 – I think a step is missing in which all media will be taken out, 40ul
of fresh media added and only then HGF and drugs are added again.

Protocol 5

• 4 – PLX4720 must be diluted to 20 mM or less before diluted into media.
This remark is true to all other planned experiments.

Protocol 6

• We used media with phenol-red for these experiments.

• 2 – On day 1 we did not change media to fresh. We only added drugs/HGF
as indicated.

• Again – Make sure that stocks of PLX4720 are not over 20 mM when added
to media.

• 3 – Cells were washed with cold PBS quickly on ice. Lysis buffer was
added to the wells on ice. Cells were scraped and cell extract moved into Eppendorf
tube.

---

## [Author Response]

*1) Two reviewers felt that more attention should be given to the*
[5]
*publication that reportedly failed to confirm a correlation between HGF
expression and outcome (Figure 3 in Straussman et al)*.

Specifically:

*a. It should be noted in the text that the same group did replicate some other
key findings of Straussman et al* – *the presence of HGF in
human melanoma tumors (in both melanoma cells and stromal cells) and the finding that
HGF is significantly enhanced in disease progression*.

We updated the language of the Introduction section to incorporate these findings by
Lezcano and colleagues.

*b. The testing for a correlation between pre-treatment HGF and clinical outcome
was done by Lezcano et al using a cohort of 23 pre-treatment samples. While we fully
support the claim by Lezcano et al that “rigorous validation studies are thus
indicated for approaches that seek to personalize such therapies to maximize
therapeutic efficacy,” we wonder if testing of 23 samples* can
*be considered as rigorous. As no power calculations are mentioned in Lezcano
et al, we would like to see some discussion of whether Lezcano et al were
sufficiently powered to make positive or negative associations. If not, how large
would the sample sizes need to be?*

While we were interested in including the experiment presented in Figure 3 in the
replication attempt, we did not because of the potentially large sample size needed.
Based on our power calculations, to be able to detect the effect size reported in [10] (*d* =
0.66609 for RAFi or RAFi + MEKi-treated patients), a total sample size of 74 would
have been needed assuming an allocation ratio of 1. However, as seen in Supplemental
Table S7, the allocation ratio was approximately 2 with an overrepresentation of
HGF-positive samples. If we were to assume the new samples had the same ratio of
HGF-negative to HGF-positive samples this would have required an increase in the needed
total sample size to 84 to account for the unequal allocation. Interestingly, Lezcano
and colleagues also saw an unequal allocation (4.75), but with an overrepresentation of
HGF-negative samples. With this unequal distribution of samples the total sample size
would have needed to be 126 to have at least 80% power. This could be implied that both
studies were underpowered to detect the effect size reported in [10], which tested 34 samples, even though both
studies were highly powered to detect larger effect sizes. However, this sort of
after-the-fact power analysis is better suited for designing a study to detect a
potential effect size, not accessing the outcome of a study. To better visualize the
range of plausible effects from each of these studies the 95% confidence interval of the
effect size is better suited. Also, using a meta-analytical approach to combine the
effect sizes provides an indication of the present knowledge of the effect. This is an
approach that will be utilized with the replication data generated from this project. We
computed the effect size and 95% confidence intervals for these studies as well as from
a meta-analytic combining the two datasets. A forest plot is included for the reviewers
to see the results visually (and will be made available on the OSF project page
(https://osf.io/p4lzc) in addition to the summary data below. Some
additional discussion is included in the manuscript.95% confidence intervalStudyEffect size (*d*)Lower limitUpper limit[10]0.66609−0.075511.39781[5]0.27606−0.808591.35422Combined studies0.54264−0.065611.15090

*c. Wilson et al (PMID: 22763448) tested the correlation between plasma HGF and
PFS/OS on 126 melanoma patients and did find a statistically significant negative
correlation that supports the findings in Straussman et al. As this is the only
available big cohort testing HGF and clinical outcome on BRAFi, this should be adding
it to the literature summary in the introduction*.

We updated the language of the Introduction section to incorporate these findings by
Wilson and colleagues.

*2) We are aware of 2 groups that have directly replicated several of
the* in vitro *experiments of the paper and have published some
results. These should be added these to the literature summary*.

*a. A group from Amgen attempted to directly replicate the key findings from
Straussman et al. Their findings* can *be found here:*
*http://cancerres.aacrjournals.org/cgi/content/meeting_abstract/73/8_MeetingAbstracts/3405**.
They show that HGF* can *rescue melanoma cell lines from BRAFi and
MEKi and that this rescue is attenuated by METi*.

We updated the language of the Introduction section to incorporate these findings by
Caenepeel and colleagues.

*b. A group from the University of Illinois was able to demonstrate that c-MET
inhibition is synergistic with BRAF inhibition in melanoma cell lines*.
*http://cancerres.aacrjournals.org/cgi/content/meeting_abstract/73/8_MeetingAbstracts/2078*

We updated the language of the Introduction section to incorporate these findings by
Etnyre and colleagues.

3) Regarding statistical power, we also have the following
suggestion:

*While it is very useful for you to leverage the previously reported effects to
compute minimum power* a priori*, what you really need is to guarantee
a minimum power on your own data. This* can *be done,* a
priori*, by including some cross-study variation. This will be helpful for you
to plan on the number of replicates and so forth. Papers by Giovanni Parmigiani and
collaborators at the Dana Farber provide some estimates about cross-study variation
that could be used for this purpose. Worst case, you should budget some additional
variability because of cross-study reproducibility, and increase the sample size as
appropriate. We also want you to compute and report power post-hoc/on-the-fly on your
own data. Some minimum power should be guaranteed using summaries of your own
data*.

We thank the reviewers for these suggestions. The cross-study variation, such as
approaches that utilize the 95% confidence interval of the effect size, can be useful in
conducting power calculations when planning adequate sample sizes for detecting the true
population effect size, which requires a range of possible observed effect sizes.
However, the Reproducibility Project: *Cancer* Biology is designed to
conduct replications that have 80% power to detect the point estimate of the originally
reported effect size. While this has the limitation of being underpowered to detect
smaller effects than what is originally reported, this standardizes the approach across
all studies to be designed to detect the originally reported effect size with at least
80% power. Also, while the minimum power guarantee is beneficial for observing a range
of possible effect sizes, the experiments in this replication, and all experiments in
the project, are designed to detect the originally reported effect size with a minimum
power of 80%. Thus, performing power calculations during or after data collection is not
necessary in this replication attempt as all studies included are already designed to
meet a minimum power or are identified beforehand as being underpowered and thus are not
included in the confirmatory analysis plan. The papers by Giovanni Parmigiani and
collaborators highlight the importance of accounting for variability that can occur
across different studies, specifically gene expression data. While it is possible for a
difference in variance between the originally reported results and the replication data
this will be reflected in the presentation of the data and a possible reason for
obtaining a different effect size estimate.

Comments on the specific protocols:

*Protocols 1 and 2 – We think that the protocols are mixed and experiment
detailed in protocol 2 should be protocol 1 and vice versa. This should be corrected.
Below we refer to the protocols as they appear in the file that we
received*.

Yes, these two are mixed. We corrected them and edited the language to better reflect
the intent of each experiment.

Protocol 1

*• When growing SK-MEL-5-GFP cells make sure that >85% of cells are
GFP labeled. If number of GFP positive cells are dropping one* can
*use FACS or antibiotics to enrich again for GFP positive cells. We did not
grow the cells under antibiotic selection on a regular basis*.

We added this note to each protocol using SK-MEL-5-GFP cells to confirm at least 85% of
the cells are GFP-positive before the start of the experiment.

*• Microplate reader used is different from original and should be labeled
with a **.

We corrected this for all protocols that use this instrument.

*• We used Corning #3712 plates and did specify that in the methods
section. Please remove the * and remove the comment: “Original
unspecified”*.

We corrected this for all protocols that use these plates.

*• 1c – as specified in the methods section we maintained cells in
DMEM from Invitrogen (#10569-010). While using phenol-red free DMEM for the
screens is a good idea (we did the same) I would recommend supplementing it with
sodium pyruvate as the DMEM that we used had Sodium pyruvate in it. When using
Phenol-red free media we used to add Sodium pyruvate from Cellgro (Cat
#25-000-CI) to a final of 1 mM*.

We included the addition of 1 mM sodium pyruvate to all medium missing this
supplement.

*• 1d – we have plated cells on 384-well plates using the Combi
cell platter (**http://www.thermo.com.cn/Resources/201306/21143420640.pdf**).
This resulted in very accurate plating. I don't know how the replicating lab is
planning to plate cells on 384-well plates. If manual plating is planned make sure
that no air bubble is present at the bottom of the well after plating as
this* can *frequently occur for those unexperienced with manual
plating of 384-well plates*.

The replicating lab will use automated methods for this protocol; specifically a Biomek
FX auto workstation. We have updated the manuscript to reflect this.

Protocol 2

*• Read GFP only after cells have completely settled down. As indicated in
the paper we used to plate cells on day 0 and read GFP for the first time on day
1*.

We corrected this in protocols 1 and 2 to indicate the plates will be read one day after
plating.

*• Read GFP from wells with media and no cells as well. Before analyzing
results subtract reading from clear–media wells from wells that have cells. We
noticed that reading from media-only wells* can *change from day from
day and thus subtract the reading from media-only wells from wells with cells. To
this end we always make sure to have media-only wells on each plate with a total
volume that is equivalent to test-wells. This remark is true to all experiments in
all protocols*.

We added the media-only wells to Protocol 2 and for all protocols analyzing
proliferation assays the subtraction of media-only wells from the wells with cells is
indicated for each GFP fluorescence reading.

Protocol 3

*• 2b – Seems like 50ul and not 60ul is a better control for the
wells that will have 20ul of cancer cells + 20ul of PCM + 10ul of
drug*.

We corrected the volume of the media-only wells to reflect the plating strategy.

*• From the protocol it seems like stromal cells are plated once. I have
plated stromal cells 3 times (each time 3 days before I needed it) to make sure that
I have fresh PCM on days 0,1 & 4*.

We updated the language to more clearly describe the strategy of generating fresh PCM on
three separate occasions.

*• This protocol involves a few cycles of media change in 384-well plates.
We have done so using a CyBi robotic liquid handler. Do the replicating lab plan to
use a robotic liquid handler? From my experience it is not easy to take out the exact
same amount of media from 384-wells manually making sure not to touch the bottom and
disturb the cells. If a robotic plate handler* can *be used I would
recommend using it, as manual handling of hundreds of 384-wells might be a source of
a lot of noise in the experiment. Lastly - both extraction and addition of liquid
from the wells should be done gently. Cells are under the treatment of BRAFi and
might be displaced more easily that non-treated cells. If using a robotic system
please do not exceed a rate of 5-10μl/sec and do not let the tip end closer
than 1 mm to the well bottom*.

As mentioned above, the replicating lab will be using an automated workstation, the
Biomek FX, to perform the liquid handling.

*• 8a – Subtract the reading from media only wells first and only
then subtract reading of day 1 from day 7. This remark is true for all
protocols*.

We added for all protocols analyzing proliferation assays the subtraction of media-only
wells from the wells with cells before normalizing to day 1 GFP fluorescence
readings.

Protocol 4

*• 1a – please correct number to match your planned 2500
cells/well*.

We have corrected the number to match the planned seeding for the experiment. Since a
ViCell XR is not being used, the number of 2500 cells/well used by Straussman and
colleagues will be increased to 2800 cells/well to better reflect the number of live
cells/well. This is the same proportional increase in cell number used in the other
experiments where 1700 cells/well is increased to 1900 cells/well.

*• 5 – I think a step is missing in which all media will be taken
out, 40ul of fresh media added and only then HGF and drugs are added
again*.

We have added this missing step to the protocol.

Protocol 5

*• 4 - PLX4720 must be diluted to 20 mM or less before diluted into media.
This remark is true to all other planned experiments*.

We corrected the dilution of PLX4720 in Protocol 5 and 6 to be 20 mM before dilution
into media.

Protocol 6

*• We used media with phenol-red for these experiments*.

We corrected the DMEM to use Sigma D6429, which contains phenol-red, L-glutamine, and
sodium pyruvate, which is the same formulation as Invitrogen 10,569-010.

*• 2 – On day 1 we did not change media to fresh. We only added
drugs/HGF as indicated*.

We corrected the language to reflect this.

*• Again – Make sure that stocks of PLX4720 are not over 20 mM when
added to media*.

We corrected the dilution of PLX4720 to be 20 mM before dilution into media.

*• 3 – Cells were washed with cold PBS quickly on ice. Lysis buffer
was added to the wells on ice. Cells were scraped and cell extract moved into
Eppendorf tube*.

We corrected the language to reflect the lysis of cells in wells on ice.

We also included some reformatting/editing of the introduction, sampling, analysis, and
power calculations sections to be more thorough and clear in our reporting. This
includes the addition of ANOVA tests that occur before the planned comparisons
(t-tests). Previously we only included the planned t-tests. Additionally, we excluded
the pMet Tyr1234/5, RAF1, and pRAF1 antibodies in Protocol 6 (Figure 4C/Supplemental
Figure 19) as these were not analyzed in the protocol and were redundant with antibodies
already described (pMet Tyr1349) and noted in the ‘Known differences from
original study section’. Also, the additional control (untreated cells in
addition to vehicle treated) that was described previously, but not included in this
‘Known differences from original study section’ is included for all
protocols.